# Amphiphysin AoRvs167-Mediated Membrane Curvature Facilitates Trap Formation, Endocytosis, and Stress Resistance in *Arthrobotrys oligospora*

**DOI:** 10.3390/pathogens11090997

**Published:** 2022-08-31

**Authors:** Peijie Cui, Mengqing Tian, Jinrong Huang, Xi Zheng, Yingqi Guo, Guohong Li, Xin Wang

**Affiliations:** 1State Key Laboratory for Conservation and Utilization of Bio-Resources in Yunnan, Yunnan University, Kunming 650091, China; 2Key Laboratory for Potato Biology of Yunnan Province, The CAAS-YNNU-YINMORE Joint Academy of Potato Science, Yunnan Normal University, Kunming 650091, China; 3Core Technology Facility of Kunming Institute of Zoology (KIZ), Chinese Academy of Sciences (CAS), Kunming 650223, China

**Keywords:** *Arthrobotrys oligospora*, *AoRvs167*, membrane bending, endocytosis, trap formation, stress resistance

## Abstract

Bin1/Amphiphysin/Rvs (BAR) domain-containing proteins mediate fundamental cellular processes, including membrane remodeling and endocytosis. Nematode-trapping (NT) fungi can differentiate to form trapping structures through highly reorganized cell membranes and walls. In this study, we identified the NT fungus *Arthrobotrys oligospora* ortholog of yeast Rvs167 and documented its involvement in membrane bending and endocytosis. We further confirmed that the deletion of *AoRvs167* makes the fungus more hypersensitive to osmotic salt (Nacl), higher temperatures (28 to 30 °C), and the cell wall perturbation agent Congo red. In addition, the disruption of *AoRvs167* reduced the trap formation capacity. Hence, *AoRvs167* may regulate fungal pathogenicity through the integrity of plasma membranes and cell walls.

## 1. Introduction

The curvature of the membrane plays an essential role in eukaryotic cell division, endo/exocytosis, and vesicular trafficking [1,2,3]. Bin/Amphiphysin/Rvs (BAR) domain-containing proteins are closely associated with the generation of curves in these biological processes [4,5]. Rvs161/167 (reduced viability upon starvation protein 161/167) is a heterodimer of the N-BAR domain family of proteins that promotes membrane curvature by binding to membranes [6,7]. It is orthologous to mammalian amphiphysin and Bin1; it has also been found to mediate vesicular formation in yeasts and filamentous fungi [8,9]. Both Rvs proteins consist of a BAR homology domain, which allows them to form functional heterodimers [10]. In the BAR domain, there is an amphipathic helix that can cause a high degree of membrane curvature [11]. Thus, Rvs161/167 can function as a sensor for membrane curvature.

Structurally, the most apparent difference between the Rvs167 and the Rvs161 protein is the presence of an SH3 domain in the C-termini, followed by a BAR domain [12]. In addition, Rvs167 contains a non-conserved GPA region (Gly-Pro-Ala-rich), which serves as a link between BARs and SH3 [13]. Numerous lines of evidence indicate that Rvs167 plays a multi-faceted role in fungi. As an example, *Magnaporthe oryzae* Rvs167 is localized to the appressorium pore and involved in the extension of the penetration peg during the pathogen invasion of the host plant [14]. In *Saccharomyces cerevisiae* and *Candida albicans*, Rvs167 has commonly been associated with endocytosis regulation, the loss of which can result in defects of vesicle trafficking, cell wall integrity, and cellular stress [15,16]. Taken further, it plays a role in fungal virulence, morphogenesis, and polarization of actin patches in *Cryptococcus neoformans* and *Candida albicans* [17,18]. Mammalian amphiphysin and Bin1 proteins, with their close functional similarity to fungi, are thought to be involved in endocytosis-mediated synaptic vesicle recycling [19,20]. It is therefore believed that Rvs167 regulates actin, endocytosis, and polarized cell growth.

There are a number of nematode trapping (NT) fungi that control soil pests [21]. They can capture, immobilize, and kill nematodes with their diverse and specific trapping organs, including adhesive networks, adhesive knobs, adhesive columns, and constricting and non-constricting rings [22,23]. *Arthrobotrys oligospora* is a ubiquitous NT fungus that is capable of controlling the nematode population in nature [24]. The organism grows as a saprophyte in the presence of rich nutrients and then as a parasite once nutrient sources are limited or when nematode prey is present [25]. For a long time, this model has been used to study mechanisms of cell type switching and interplay between predacious fungi and their hosts [26]. Many trap regulators have been found to be involved in endocytosis and autophagy, including Rab GTPase (AoRab-7A), Arf-GAP Glo3 (AoGlo3), mitogen-activated protein kinase AoSlt2, and calcium/calmodulin-dependent protein kinases [27,28,29,30,31]. Furthermore, arrestin proteins may be influenced by the endocytosis pathway when it comes to fungal trap development and virulence against nematodes [32]. We previously demonstrated that bacteria can trigger NT fungi to switch lifestyles by secreting urea. The downstream molecule ammonia also contributes to trap formation [25]. A similar effect of ammonia on endocytosis has also been observed in human cells and other fungi [33,34].

In this study, we aim to examine the functions of yeast ortholog Rvs167 (*AoRvs167*) in *A. oligospora*, which has been shown to regulate the development of trap devices and pathogenicity against nematodes, control membrane curvature, and endocytosis and to allow for stress resistance to *A. oligospora*.

## 2. Results

### 2.1. Sequence and Protein Structure Analysis of AoRvs167

First, we retrieved the sequences of AoRvs161 (AOL_s00215g615, Gene ID: 22899063) and AoRvs167 (AOL_s00075g198, Gene ID: 22891708) from NCBI; AoRvs161 contains 261 amino acids in total, and AoRvs167 has 354 amino acids. Both proteins contain a general N-BAR domain. AoRvs167 typically presents the same structure as Rvs167 proteins from fungi, which also includes a glycine–proline–alanine-rich region associated with the SH3 domain at the C-terminus (Figure 1A). Based on STRING, AlphaFold2, ClusPro2.0, and molecular dynamics simulation, we found that AoRvs161 and AoRvs167 can interact (Figure 1B, Appendix A) and form a stable dimer complex (Figure 1C, Appendix A). Furthermore, it was confirmed by the phylogenetic tree analysis that Rvs proteins are conserved among eukaryotic organisms (Figure 1D).

### 2.2. AoRvs167 Is Associated with the Colony Growth 

The NT fungus *A. oligospora* has previously been shown to use ammonia as a signaling molecule against trap formation [33]. *AoRvs161* (Gene ID: 22899063) and *AoRvs167* (Gene ID: 22891708) expression profiles were enhanced by 3.0617 and 2.6041 when exposed to ammonia (Appendix A). To identify Rvs functions, we deleted the coding gene of *AoRvs167* from *A. oligospora* (Figure 2A, Appendix A). On CMY medium, the WT strain and the mutant strain Δ*AoRvs167* were cultured for observation of sporulation; the mutant had 39.7% fewer spores than the WT strain (Figure 2B). Δ*AoRvs167* had similar colony phenotypes to the WT strain in TGA and TYGA, but the growth was slowed by nearly 30% in PDA at 28 °C (Figure 2C). In comparison to the WT and mutant strains, there were no significant differences in the number of mycelial septa (Appendix A). Therefore, *AoRvs167* regulates growth and sporulation in *A. oligospora*. 

### 2.3. AoRvs167 Mediates Membrane Curvature and Endocytosis

As scaffolding modules, BAR domains bend and remodel membranes by binding to the plasma membrane (PM). Therefore, we focused on the morphological difference of the PM. A significant observation from the SEM images was that the mutant cell wall phenotype showed obvious wrinkles as compared to the WT. Additionally, TEM images revealed that the mutant PM was straightened and tense at the septum and the PM region. In contrast, the WT membrane exhibited some bending, so the membrane structure seemed more flexible. (Figure 3A,B, Appendix A). Subsequently, we wondered whether the *AoRvs167* mutation would affect the PM endocytosis response to FM4-64, which is an endocytic marker for filamentous fungi. When stained for 1 min, the mutant cells displayed a faint red signal; however, the red fluorescent was largely absorbed by the WT hyphal cells. A significant number of dye puncta could be seen in the WT hyphae after staining for 10 min, but only a small number could be seen at the mutant PM (Figure 3C). By comparing the fluorescence intensities, the results were generally consistent with the result reflected in Figure 3C (Appendix A). Moreover, qRT-PCR revealed that the mutation of *AoRvs167* suppressed endosome formation-related genes (*vps4*, *snf7*, and *vps28*) (Figure 3D, Appendix A). Based on these observations, the integrity of the cell wall and PM was disrupted, and the endocytosis response was diminished in *A. oligospora*. 

### 2.4. AoRvs167 Deletion Significantly Suppressed the Trap Generation 

Adhesive networks are typical markers of the pathogenicity-related switch of hyphae in *A. oligospora*. Using ammonia and nematodes as induction substances, we tested the effects of the *AoRvs167* gene on trap formation. Δ*AoRvs167* had almost no adhesive network after nematodes or urea were induced (Figure 4A,B). As a consequence, nematode animals could not be killed by the mutant strain in the test (Figure 4C,D). Hence, AoRvs167 acts as a robust regulator of trap structures in *A. oligospora*.

### 2.5. Rvs167 Gene Regulates Stress Resistance of A. oligospora

For testing the effects of *AoRvs167* on stress tolerances, osmotic salts (NaCl, 0.1 to 0.3 M), cell wall perturbing agents (Congo red, 0.05 to 0.15 mg/mL; Calcofluor white, 50 to 200 μg/mL), and temperature ranges (20 to 30 °C) were used. In contrast to the WT strain, the mutant strain was more sensitive to NaCl, Congo red, and higher temperatures (Figure 5A). In addition, RGI (%) is used to illustrate the stress levels of WT and mutant strains. RGI values were 36.8% and 79.2% for *AoRvs167* and WT strains, respectively, with NaCl (0.2 M). Especially in 0.3M NaCl, the mutant strain reached 96.4% of the RGI and almost stopped growing (Figure 5B). When treated with Congo red, the RGI were 28.3% and 77.7% in the WT and *AoRvs167* strains, respectively. Furthermore, the mutant colony nearly failed to expand with the RGI of 93.4% (Figure 5C). In addition, CFW analysis further confirmed that cell wall integrity had been interrupted by *AoRvs167* loss in *A. oligospora* (Appendix A). Finally, for the temperature effect on the growth of the mutant, we found that Δ*AoRvs167* grew normally at 20 and 25°C but displayed more severe growth defects at 28–30°C (Figure 5D). As a result, *AoRvs167* plays an essential role in adapting *A. oligospora* to external factors.

## 3. Discussion

NT traps are highly specified structures that are distinct from vegetative hyphae in NT fungal groups [21]. Morphologically, the development of trapping structures is linked to the apparent reorganization of the plasma membrane (PM) and cell wall (CW). PM is the frontier between the inside and the outside of the cells. Membrane composition is controlled by releasing endocytic vesicles, which is a vital dynamic cellular process mediated by the Rvs161/Rvs167 proteins [35]. Our study identified the key functions of the *A. oligospora* protein AoRvs167, the loss of which led to a lag in the endocytosis response against FM4-64 dye, that was attributed to a lack of membrane elasticity (Figure 3C). Recent studies have shown that *S. cerevisiae* Rvs167 is ubiquitinated in the presence of a high-quality nitrogen source, thereby promoting endocytosis [36]. In *A. oligospora*, the WT hyphae can form large numbers of traps against the exposures of ammonia or nematodes, which may be due to promoting endocytosis. Parallel to this, the deletion of *AoRvs167* may suppress endocytosis and inhibit trap formation. In NT fungi and other species, ammonia can cause an increase in endocytosis, so we hypothesized that reduced endocytosis would explain poor trap formation. Another study found that arrestin scaffolding proteins regulate partial trap formation through endocytosis. Several endocytic proteins, including Sla1 and Rvs167, have been identified as targets of Rsp5 [13]. In yeast, arrestin proteins such as ART3 and ART4 can recruit these proteins. Thus, the phenotypic defects caused by AoRvs167 deletion might be the result of impaired endocytosis processes in *A. oligospora*. 

Furthermore, the PM are also involved in CW biosynthesis. Therefore, it is possible that an abnormal PM may be unable to cope with environmental stresses, such as elevated osmotic pressure, high temperatures, etc. Lack of Rvs167 in yeast results in increased sensitivity to high temperature, Congo red, and salt stress [37,38]. Similar to yeast, the mutant also became more sensitive to these stressors, probably because the membrane was not bending or fluid. In yeast, Congo red may alter the call wall integrity by targeting chitin synthesis [39]. The test of *AoRvs167* mutant colonies against Congo red also revealed a sharp decrease in resistance to the cellular wall perturbations. As a result, fungal cell membranes may recruit AoRvs proteins to regulate stress when communicating with external environments. 

Finally, *A. oligospora* contains both yeast orthologs Rvs161 and Rvs167; however, only *AoRvs167* was shown and characterized in this study. This study lacks data on the detailed function of *AoRvs161*, even though evidence suggests they have similar roles to *AoRvs167* in endocytosis. The *AoRvs161* (AOL_s00215g615) and *AoRvs167* (AOL_s00075g198) present similar amino acid sequence to each other (48%), based on which we can suspect the functions of *AoRvs161*. In yeasts, such as *S. cerevisiae* and *C. albicans*, Rvs161 has been identified as similar to Rvs167 and shows significant defects in endocytosis, actin organization, and virulence. Furthermore, Rvs161 and Rvs167 form an obligatory heterodimer through the N-terminus [40]. *AoRvs161* and *AoRvs167* genes were up-regulated at the same level, especially when ammonia was induced. Accordingly, we speculated that *AoRvs161* mutations would cause similar phenotypes to those of *AoRvs167* regarding endocytosis regulation, PM morphogenesis, stress resistance, and pathogenicity in NT fungi.

In summary, this study indicates that the loss of *AoRvs167* is associated with multiple fundamental phenotypes, such as PM bending, delay in endocytosis, salt resistance, cell wall anti-disturbance, and even NT fungi morphogenesis and their virulence against nematodes (Figure 6). Due to the fact that Rvs167 is defined as a membrane endocytic protein in some studies, this research adds a new understanding of how membrane structure may relate to NT fungal stress resistance and pathogenicity.

## 4. Materials and Methods

### 4.1. Fungal Strains and Culture Conditions

*Arthrobotrys oligospora* ATCC 24927 is grown on PDA (20% potato and 2% dextrose) at 28 °C in the State Key Laboratory for Conservation and Utilization of Bio-Resources in Yunnan. *S. cerevisiae* FY834 was cultured in YPD broth (1% yeast, 2% peptone, and 2% dextrose) or on a solid YPD (YPD with 1.5% agar), and the recombinant yeasts were selected with SC-Ura media [41]. *Escherichia coli* strain DH5α (TaKaRa Bio, Shiga, Japan) was used as a host for plasmids pRS426 and pCSN44. Furthermore, liquid TG (1% tryptone and 1% glucose) medium was used for mycelium collection; a PDAS (PDA supplemented with 10 g/L molasses and 0.4 M saccharose) medium was used for protoplast regeneration, solid CMY (2% maize boiled for more than half an hour, 0.5% yeast extract, and 1.5% agar) was used for the conidia culture, TYGA (1% tryptone, 0.5% yeast extract, 1% glucose, 0.5% molasses, and 1.5% agar), PDA, and solid TG (TG containing 1.5% agar) were used for the growth rate comparison of the WT and mutant strains; WA (deionized water with 1.5% agar) was used for trap tests. *Caenorhabditis elegans* was cultured at 26 °C in an oatmeal water medium.

### 4.2. Sequence and Phylogenetic Analyses of AoRvs167

The amino acid sequences of *AoRvs161* and *AoRvs167* were extracted from NCBI GenBank (https://www.ncbi.nlm.nih.gov/genbank/, accessed on 14 August 2022). The protein interaction model was predicted using STRING v11 [42]. The 3d structure of proteins was determined by AlphaFold2, a website for protein prediction (https://colab.research.google.com/github/sokrypton/ColabFold/blob/main/AlphaFold2.ipynb, accessed on 14 August 2022) [43]. After entering the amino acid sequence, we ran the code to make predictions using default parameters. In the prediction program, there are five parts: Protein sequence input, MSA options, Advanced settings, Run Prediction, and Display 3D Structure. With ClusPro2.0, we docked the two proteins, Rvs167 as receptor and Rvs161 as ligand; uploaded the protein PDB predicted by AlphaFold2; and then selected the first (best) model in the output result for display [44,45,46,47]. To verify the stability of the complex, we used Gromacs to perform a 10 ns molecular dynamics simulation of the complex [48,49], and PyMOL (V 2.5 Open-Source) was used to display the protein structure [50]. The specific phylogenetic tree analysis method is as follows: To begin, we obtained amino acid sequences from NCBI and imported them into MEGA for multiple sequence comparison (Align by ClustalW), followed by trimming the sequence at the beginning and end, as necessary. Our next step was to use MEGA to find the best protein model. By analyzing the BIC, AICc, and InL values, we determined that LG+G is the best evolutionary tree model (Appendix A). Lastly, we used Maximum Likelihood to perform an evolutionary analysis [51,52,53].

### 4.3. Deletion of the AoRvs167

In a homologous recombination, the target genes were disrupted [41]. *AoRvs167*_5F/*AoRvs167*_5R, *AoRvs167*_3F/*AoRvs167*_3R were used as PCR primer pairs with *A. oligospora* genomic DNA as the template, and the hygromycin B resistance cassette (*hph* cassette) was amplified using hph-F/hph-R from plasmid pCSN44. Together with linearized vector pRS426 (digested with *EcoRI*/*XhoI*), the three fragments were transformed into *Saccharomyces cerevisiae* strain FY834 cells to create pRS426-gene-hph. PCR primers 5F/3R were used to amplify the deletion cassette. Using a PEG-mediated approach, the purified deletion fragments were transformed into protoplasts. The mutant colonies were selected on PDA medium containing 200 mg/mL hygromycin B, and PCR amplification was performed with specific primers (S) [54]. As further verification, RT-PCR was conducted on mutant strains (Appendix A).

### 4.4. Comparison of Mycelial Growth, Sporulation and Hyphal Morphology

Initially, the WT strain was cultured on PDA at 28 °C for 6 days, and then, the same size (6 mm in diameter) fungus blocks were incubated on PDA, TG, or TYGA plates at 28 °C for 6 days. To assess mycelial growth, colony diameters were measured. Fungal hyphae septa were stained with 50 μg/mL calcofluor white (Sigma-Aldrich, St. Louis, MO, USA) and observed under a Nikon inverted fluorescence microscope. Then, we quantified the number of septa per hyphae (50 per colony), in triplicate, and statistically analyzed it. We cultured WT and mutant strains on CMY medium at 28 °C for 5 days, and 8 mL of water was used to prepare a conidial suspension, from which 1 mL was counted.

### 4.5. Endocytosis Analyses

We cultured wild-type and mutant strains on a cellophane-coated water agar (WA) medium for 3 days. The cellophane layer of the medium was cut, stained with FM4-64 at a final concentration of 5 μg/mL (Biotium, Fremont, CA, USA) for 1 min and 10 min, and rinsed with ddH_2_O three times. Stained samples were observed under a fluorescence microscope (Nikon, Tokyo, Japan) [28]. We used image J (1.53q) to analyze the fluorescence intensity of FM4-64 staining [55].

### 4.6. TEM and SEM Sample Preparation

TEM sample preparation: WT and mutant strains were grown for 7 days on PDA solid media, inoculated into PDA liquid medium, cultured at 28 °C for 2 days, filtered through a funnel, and washed twice with PBS, and the mycelium was then put into a tube of 1.5 mL (fixed with 1.5% glutaraldehyde) and shaken to evenly distribute them. Mycelia were stored at 4 °C after collection. Afterward, it was sectioned and observed under transmission electron microscopy. For SEM, samples were fixated with glutaraldehyde for 30 min; dehydrated in grades of ethanol (30%, 50%, 70%, 80%, and 90%; each step took 15 min); and then immersed successively in 100% ethanol for 15 min each, in ethanol: isoamyl acetate liquor (1:1, *v*/*v*) liquor for 10 min each, and in 100% isoamyl acetate for 10 min each. Afterward, samples were transferred to a critical point dryer using liquefied carbon dioxide as a transition fluid [56].

### 4.7. Trap Formation and Pathogenicity Assays

Conidia were collected from WT and mutant strains cultured on CMY medium at 28 °C for 10 days, and 3 × 10^3^ conidia of the WT and mutant strains were spread on WA plates (6cm) and cultured at 28 °C for 48 h. We then added 200–300 nematodes to each plate to induce trap formation, and we observed and recorded the number of traps at 24 h. In the same manner, we added 1 mL of ammonia to each plate to induce trap formation at 28 °C, and we observed and recorded the number of traps at 72 h.

### 4.8. Stress Resistance

With a PDA solid medium as the base medium, a high-salt medium and a cell wall perturbation medium were prepared. NaCl concentrations ranged from 0.1 to 0.3 M, Congo red concentrations ranged from 0.05 to 0.15 mg/mL, and temperatures ranged from 20 to 30 °C. Initially, 6 mm colonies were formed by inoculating WT and Rvs167 mutant strains simultaneously. After growing at 28 °C for 6 days, we measured the colony area and calculated RGI. In addition, Calcofluor white concentrations ranged from 50 to 200 μg/mL, and strains were cultured at 25 °C for 5 days. RGI:(Sc − St)/(Sc − d) × 100, where Sc and St represent the colony area of the control group and the experimental group, respectively, and d is the colony diameter of the initial area [29].

### 4.9. Statistical Analyses

We used GraphPad Prism version 8.3.0 (GraphPad Software, San Diego, CA, USA) for graphing and data analysis, with three replicates per experiment, and data are standard deviation (SD) of the mean. Unpaired *t* test, Multiple *t* test and Two-way ANOVA were applied. * *p* < 0.05, ** *p* < 0.01, *** *p* < 0.001, **** *p* < 0.0001.

## Figures and Tables

**Figure 1 pathogens-11-00997-f001:**
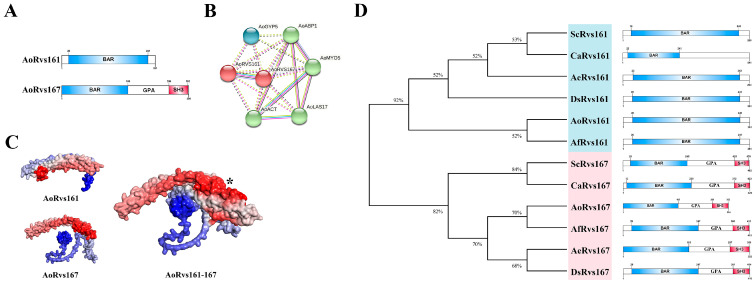
Bioinformatics analysis of the Rvs proteins. (**A**) The domain predictions of AoRvs161 and AoRvs167 proteins; the numbers represent the lengths of the proteins. (**B**) The prediction of the protein interaction network between AoRvs161 and AoRvs167 by STRING. (**C**) AoRvs161 and AoRvs167 protein structures were predicted by AlphaFold2, and the protein complex model was predicted by ClusPro2.0; asterisks indicate possible binding sites between the two proteins. (**D**) The phylogenetic tree of Rvs in fungi; the phylogenetic tree was drawn using Mega 7 software.

**Figure 2 pathogens-11-00997-f002:**
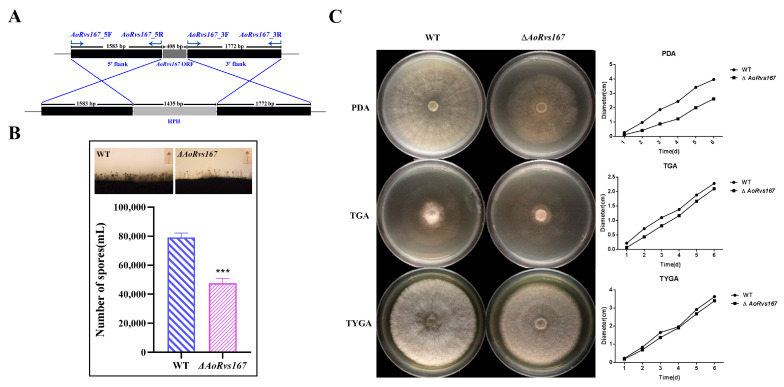
Mycelial growth and conidia production between WT and Δ*AoRvs167*. (**A**) Schematic diagram of *AoRvs167* gene knockout by homologous recombination. (**B**) Conidia production in the wild type and Δ*AoRvs167*. (**C**) Mycelial growth in the WT and *AoRvs167* mutants on PDA, TGA, or TYGA media for 6d; unpaired *t* test, parametric test, *** *p* < 0.001.

**Figure 3 pathogens-11-00997-f003:**
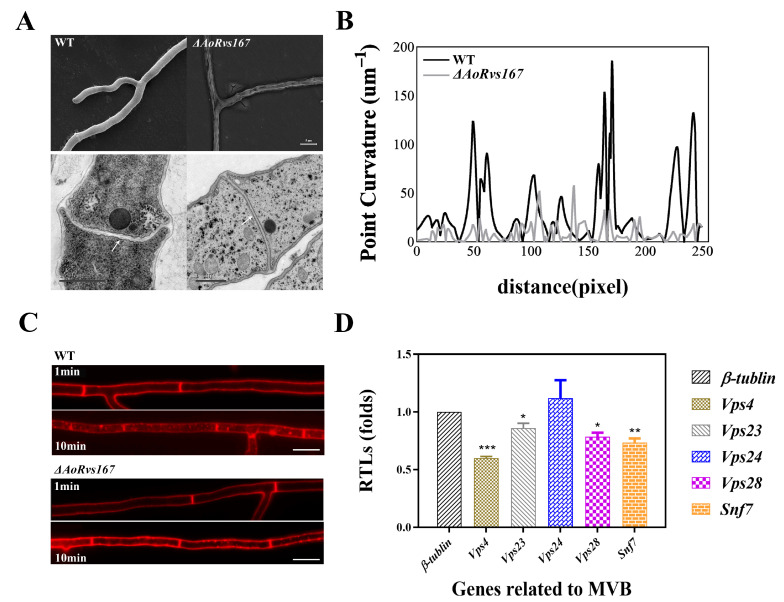
Endocytosis and membrane curvature are mediated by *AoRvs167*. (**A**) Differentiation in cell wall and membrane morphologies between the WT and Δ*AoRvs167* by TEM (Bar = 1 μm) and SEM (Bar = 5 μm), arrows indicate different degrees of curvature of the septum between the WT and Δ*AoRvs167*. (**B**) Membrane curvature analysis by 250 pixels. (**C**) Endocytosis analysis using FM4−64 staining at 1 and 10 min (Bar = 10 μm). (**D**) qRT−PCR analysis of endosome formation−related genes in mutant strains and WT. Unpaired *t* test, parametric test, * *p* < 0.05, ** *p* < 0.01, *** *p* < 0.001.

**Figure 4 pathogens-11-00997-f004:**
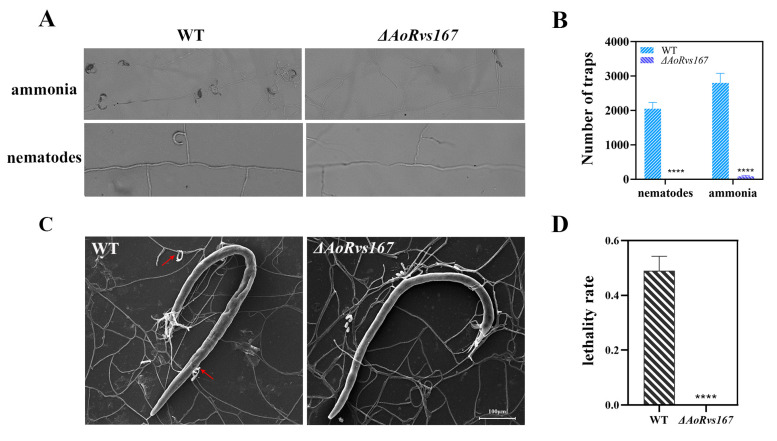
Comparison of trap formation and lethality rates in WT and Δ*AoRvs167*. (**A**) Trap formation in the WT and Δ*AoRvs167* after ammonia and nematode induction; (**B**) trap numbers in the WT and Δ*AoRvs167* after ammonia and nematode induction; Multiple *t* test, **** *p* < 0.0001. (**C**) SEM images of the WT and Δ*AoRvs167* under nematode induction conditions, red arrows represent the trap; (**D**) Lethality rate of the WT and Δ*AoRvs167* against the nematode. Unpaired *t* test, parametric test, **** *p* < 0.0001.

**Figure 5 pathogens-11-00997-f005:**
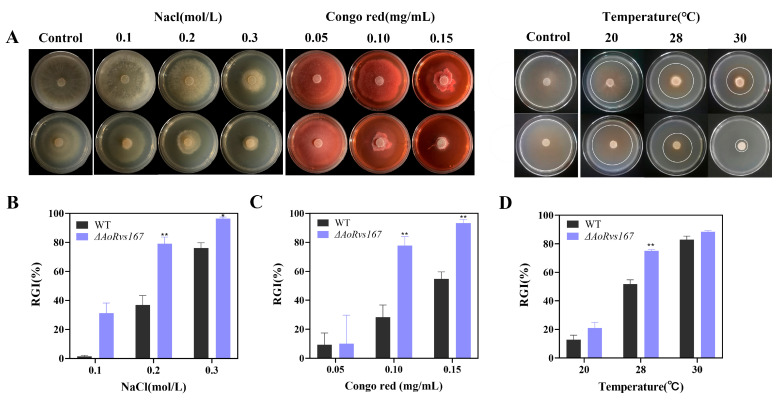
Comparison of resistance of the WT and Δ*AoRvs167*. (**A**) Colony growth of the WT and Δ*AoRvs167* against NaCl, Congo red, and temperatures. (**B**) Relative growth inhibition of the WT and Δ*AoRvs167* on NaCl. (**C**) Relative growth inhibition (RGI) of the WT and Δ*AoRvs167* on Congo red. (**D**) Relative growth inhibition (RGI) of the WT and Δ*AoRvs167* against different temperatures; the control temperature was 25 °C. Two-way ANOVA, * *p* < 0.05, ** *p* < 0.01.

**Figure 6 pathogens-11-00997-f006:**
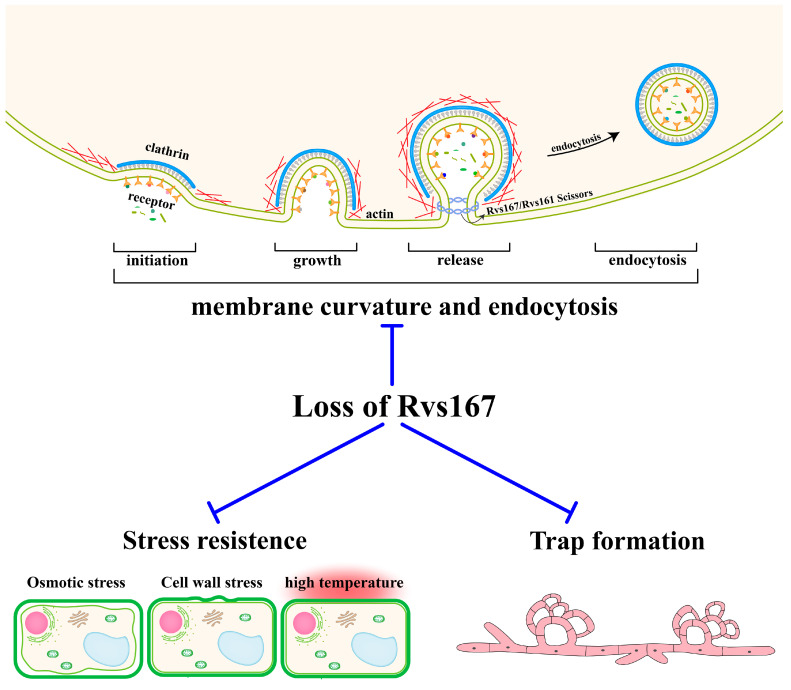
The regulatory model of the *AoRvs167* gene in *A. oligospora*. Membrane curvature is generated throughout the endocytosis process, including the initiation, growth, and release of vesicles. Deletion of the *AoRvs167* affects a range of biological processes, including PM bending, endocytosis delay, stress resistances, and trap formation.

## Data Availability

The original contributions presented in the study are included in the article/Appendix A, and further inquiries can be directed to the corresponding author.

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
