# Peer review of "Amphiphysin AoRvs167-Mediated Membrane Curvature Facilitates Trap Formation, Endocytosis, and Stress Resistance in Arthrobotrys oligospora"

_pathogens, 2022, doi:10.3390/pathogens11090997_

Round 1

Reviewer 1 Report

The authors did a good job. I recommend it in present form.

Author Response

Thank you for your approval.

Reviewer 2 Report

The manuscript was substantially improved. Only a minor typo should be corrected. In caption of Figure 5, replace ANONA by ANOVA.

Author Response

Only a minor typo should be corrected. In caption of Figure 5, replace ANONA by ANOVA.

Thanks for the correction, we have replaced ANONA with ANOVA in Figure 5 and Method 4.9.

Reviewer 3 Report

After analyzing the manuscript, it is clear that the authors made significant additions and corrections in almost all parts of the article. Now the article looks different, and the improvement in the presentation of the work is noticeably visible.

Author Response

Thank you for your approval.